# MOTION-AWARE TRANSFORMER FOR MULTI-OBJECT TRACKING

## ABSTRACT

Multi-object tracking (MOT) in videos remains challenging due to complex object motions and crowded scenes. Recent DETR-based frameworks offer end-to-end solutions but typically process detection and tracking queries jointly within a single Transformer Decoder layer, leading to conflicts and degraded association accuracy. We introduce the Motion-Aware Transformer (MATR), which explicitly predicts object movements across frames to update track queries in advance. By reducing query collisions, MATR enables more consistent training and improves both detection and association. Extensive experiments on DanceTrack, SportsMOT, and BDD100k show that MATR delivers significant gains across standard metrics. On DanceTrack, MATR improves HOTA by more than 9 points over MOTR without additional data and reaches a new state-of-the-art score of 71.3 with supplementary data. MATR also achieves state-of-the-art results on SportsMOT (72.2 HOTA) and BDD100k (54.7 mTETA, 41.6 mHOTA) without relying on external datasets. These results demonstrate that explicitly modeling motion within end-to-end Transformers offers a simple yet highly effective approach to advancing multi-object tracking.

## 1 INTRODUCTION

Multi-object tracking (MOT) aims to detect objects and maintain their identities across frames in a video. Recent advances in query-based detection have reshaped this task. The introduction of DETR Carion et al. (2020) marked a turning point, as it replaced handcrafted post-processing with an end-to-end formulation. Building on this paradigm, TrackFormer Meinhardt et al. (2022) extended DETR to MOT by propagating detection queries forward as track queries, and concatenating them with newly initialized detection queries for subsequent frames. This formulation enabled consistent learning of track queries without manual association rules. Later, MOTR Zeng et al. (2022) introduced a Temporal Aggregation Network (TAN) to update track queries between frames, alleviating issues of overfitting and improving performance on challenging datasets such as Dance-Track. These advances have established query propagation as a dominant strategy in end-to-end MOT. However, a key limitation remains: existing approaches typically process detection and track queries simultaneously within a single Transformer Decoder layer. This design introduces what we call *query collisions*. Track queries are required to follow the same object consistently across frames, whereas detection queries are reassigned at each frame through Hungarian matching. When a track query drifts from its ground-truth location, Hungarian matching may assign it to a different object that happens to be closer, resulting in identity switches and unstable training. At the same time, detection queries suffer from noisy gradients caused by drifting track queries, further reducing association performance. Figure 1b illustrates a typical collision scenario: a track query with low overlap with its true object is instead matched to a nearby distractor, while the correct detection query is left unmatched. Such conflicts degrade both training and inference performance, particularly in crowded or fast-moving scenes. To overcome these limitations, we propose the **Motion-Aware Transformer (MATR)**. MATR explicitly predicts the motion of track queries across frames, updating both their features and positional embeddings before they enter the Transformer Decoder. By anticipating object movements, MATR reduces the gap between track queries and their ground-truth targets, thereby minimizing query collisions and aligning the training process more closely with inference behavior. This motion-aware update provides consistent optimization for both detection and tracking queries, leading to substantial gains in association accuracy without adding complex

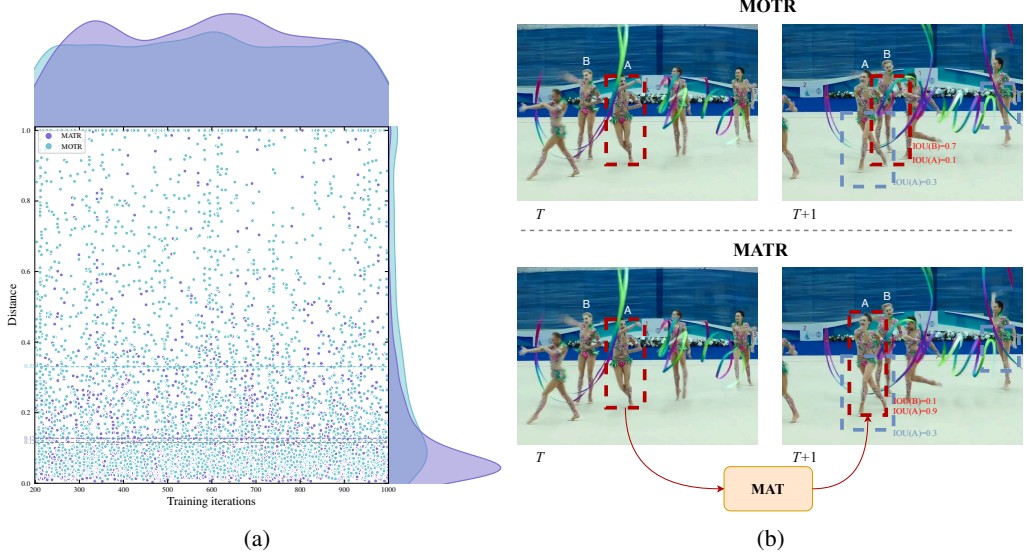

(a)                                                    (b)

Figure 1: (**a**) Distribution of track query distances ($1 - $IoU) for different methods. MATR (purple) concentrates track queries closer to ground truth compared to MOTR (cyan). (**b**) Illustration of query collisions in MOTR: a drifting track query (red) is wrongly assigned to object B under Hungarian matching, while MATR predicts the future position and keeps the track consistent.

components. As shown in Figure 1a, MATR significantly lowers the distance between queries and their assigned objects compared to MOTR, confirming the effectiveness of motion-aware prediction. In summary, MATR addresses the overlooked issue of query collisions in DETR-style MOT by introducing explicit motion prediction. This simple yet effective design improves association accuracy across diverse scenarios, leading to state-of-the-art performance while maintaining the elegance and efficiency of an end-to-end Transformer-based tracker.

## 2 RELATED WORK

### 2.1 TRACKING-BY-DETECTION

Tracking-by-Detection remains the most common paradigm for multi-object tracking. These methods typically adopt a two-step pipeline: first detecting bounding boxes in each frame, then associating detections across frames to construct object trajectories. Performance is therefore heavily dependent on the accuracy of the detection stage. SORT Bewley et al. (2016) exemplifies this approach by using the Hungarian algorithm (Kuhn, 1955) for association and a Kalman filter (Welch & Bishop, 1995) to predict object positions. Detections are matched to predicted boxes using intersection-over-union (IoU) scores. DeepSORT Wojke et al. (2017) extends SORT by incorporating appearance features extracted by a learned embedding network, which are compared via cosine distance to improve association robustness. Several methods integrate detection and re-identification more tightly. JDE Wang et al. (2020), FairMOT Zhang et al. (2021), and Unicorn Yan et al. (2022) jointly train detection and appearance embeddings in a unified framework. ByteTrack Zhang et al. (2022), based on a YOLOX detector, achieves state-of-the-art performance by considering both high- and low-confidence detections, improving recall during association. BoT-SORT Aharon et al. (2022) further enhances SORT by refining the Kalman filter state, introducing camera-motion compensation, and integrating Re-ID features. Recent efforts incorporate transformers into the association stage. TransMOT Chu et al. (2023) and GTR Zhou et al. (2022) use spatial-temporal transformers to model interactions between instances and aggregate historical context. OC-SORT Cao et al. (2023) improves upon SORT by re-identifying and rehabilitating lost tracks, further demonstrating the importance of temporal reasoning in tracking-by-detection systems.

### 2.2 QUERY PROPAGATION

The emergence of DETR-family models has popularized query propagation as an alternative, end-to-end approach for multi-object tracking. These methods maintain query embeddings across frames to track individual objects, eliminating the need for handcrafted post-processing. Track-

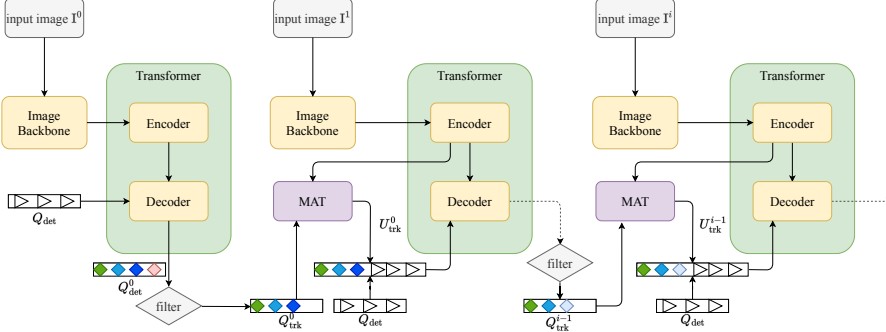

Figure 2: Overview of the MATR architecture. Unlike MOTR, MATR leverages features from the Transformer Encoder ("Memory") to update track queries $U_{\text{trk}}^i$ at time $i$. The input frame is denoted by $I^i$, and the sequence of frames by $S$. $Q_{\text{det}}$ represents learnable detection queries, and $Q_{\text{trk}}^i$ are the tracking results at time $i$. The Motion-Aware Transformer (MAT) module is supervised by a motion loss and trained jointly with the Decoder outputs. By predicting future positions, MAT updates both track query features and positional embeddings in advance.

tor++ Bergmann et al. (2019) pioneered this idea by leveraging an R-CNN Girshick et al. (2014) regression head to re-localize bounding boxes across frames. TrackFormer Meinhardt et al. (2022) extended Deformable DETR Zhu et al. (2021) to propagate detection queries as track queries, while MOTR Zeng et al. (2022) introduced a Temporal Aggregation Network (TAN) to continuously update track queries, achieving strong results on DanceTrack. MeMOT Cai et al. (2022) improved query propagation by employing both short- and long-term memory banks, while TransTrack (Sun et al., 2020) propagated queries across frames for continuous detection. Other extensions include P3AFormer Zhao et al. (2022), which integrates optical-flow-guided feature propagation, and CO-MOT (Yan et al., 2023), which introduces one-to-set label assignment and a COopetition Label Assignment (COLA) strategy to enhance detection performance. MeMOTR Gao & Wang (2023) further improved association by incorporating historical frame information. MOTRv2 Zhang et al. (2023) integrated a YOLOX detector to strengthen detection with external proposals, but at the cost of additional CNN components, making it no longer a fully end-to-end solution. MOTRv3 Yu et al. (2023) introduced several strategies, including Release-Fetch Supervision (RFS) to mitigate label insufficiency, Pseudo Label Distillation (PLD) from a pretrained 2D detector, and Query Grouping with Disturbance (TGD). Although effective, these designs add complexity and yield only modest improvements (e.g., +0.5 HOTA over MOTRv2). We argue that such strategies deviate from the original motivation of a simple, elegant end-to-end solution. Importantly, none of these methods directly address the issue of *query collisions* between track and detection queries. We empirically demonstrate their impact in Figure 1a, which shows the distance distribution between queries and ground-truth boxes. MATR (**ours**, purple) concentrates track queries near ground truth by pre-moving them before decoding, while MOTR (cyan) exhibits a more dispersed distribution, often with large deviations (mean distance 0.33 vs. 0.12 for Hungarian matching). Notably, many MOTR points cluster at $distance = 1$ (IoU = 0), highlighting its difficulty in handling large motions between frames. Although MATR reduces these cases significantly, its mean distance (0.13) still slightly exceeds the ideal Hungarian matching baseline (0.12). This analysis confirms that query collisions exist and materially degrade performance: track queries risk being matched to nearby distractors, while detection queries are corrupted by noisy track predictions. Qualitative examples in Figure 5 further illustrate how such collisions affect association in crowded scenes.

## 3 PROPOSED APPROACH

### 3.1 BASELINE IMPROVEMENT

Since the introduction of MOTR, which was built on Deformable DETR Zhu et al. (2021), newer DETR-family frameworks such as DAB-Deformable DETR Liu et al. (2022) have demonstrated improved detection performance. To establish a stronger baseline, we adopt the bounding box propagation mechanism from DAB-DETR, where bounding boxes are propagated as positional encodings for queries. Specifically, bounding boxes are initialized randomly as $N_{\text{det}} \times [x_0, y_0, h_0, w_0]$, where $N_{\text{det}}$ is the number of detection queries, and iteratively refined by successive decoder layers as $N_{\text{det}} \times [\Delta x, \Delta y, \Delta h, \Delta w]$. It is important to note that we use only the box propagation strategy from Zhu et al. (2021), rather than adopting the full DAB model. Fully incorporating DAB increases

parameters from 42M to 49M, but unexpectedly reduces HOTA from 69.4 to 67.6. Unlike MOTR, which gradually increases sequence length during training to address convergence challenges, we maintain a fixed sequence length $S$ without stability issues. For data augmentation, we simulate object entry and exit by randomly dropping track queries from previous frames, rather than artificially introducing new data. This design allows detection queries to rediscover dropped objects as newly appearing in later frames. To ensure a fair comparison with MOTR, we do not use pretrained weights from DAB-DETR. As reported in Table 1a, this strengthened baseline surpasses MOTR by approximately 5 points, providing a solid foundation for evaluating our proposed method.

## 3.2 METHOD OVERVIEW

We propose the **Motion-Aware Transformer (MAT)** module, which explicitly captures the motion information of each track query by predicting its future position. The overall framework is illustrated in Figure 2. Similar to many DETR-family methods, we employ an image backbone and a Deformable Transformer Encoder to extract features from the input frame $I$. For the first frame, detection queries $Q_{\text{det}}$ and positional embeddings PE are randomly initialized. The query tensor $Q_{\text{det}}$ has shape $N_{\text{det}} \times D$, where $D$ is the embedding dimension, and PE has shape $N_{\text{det}} \times 4$, corresponding to bounding boxes $[x_0, y_0, h_0, w_0]$. A sinusoidal encoding is applied to PE to produce positional embeddings $E_{\text{det}}$ of the same dimension as $Q_{\text{det}}$. The sum $Q_{\text{det}} + E_{\text{det}}$ is passed through the Deformable Transformer Decoder to generate the initial detection results $Q_{\text{det}}^0$. These detections are filtered using an IoU threshold to obtain high-quality track queries $Q_{\text{trk}}^0$. For subsequent frames, given the track queries from the previous time step, $Q_{\text{trk}}^{i-1}$, the MAT module leverages the "memory" features extracted by the Transformer Encoder from the current image $I^i$. Using these features, MAT predicts updated positions and refines query embeddings, producing the updated track queries $U_{\text{trk}}^{i-1}$. These updated track queries are then fed into the Deformable Transformer Decoder, which generates the final detection results $Q_{\text{det}}^i$. In this way, MAT anticipates object motion across frames, reduces query drift, and improves consistency between detection and tracking queries.

## 3.3 MOTION-AWARE TRANSFORMER

Unlike prior works such as MOTR Zeng et al. (2022) and MOTRv2 Zhang et al. (2023), which update track query features solely through self-attention, we argue that this design suffers from three key limitations: 1) without explicit supervision and guidance, track queries do not learn motion effectively; 2) self-attention alone fails to properly integrate the previous track queries $Q_{\text{trk}}^{i-1}$ with features from the current frame, which negatively impacts the subsequent Transformer Decoder; and 3) leaving positional embeddings unchanged can cause a mismatch between feature and positional information, further impairing training. Motivated by these issues, we propose the **Motion-aware Transformer (MAT)** module. As illustrated in Figure 4, MAT captures the motion trajectory of each track query by explicitly predicting its future position. This is achieved by extracting *memory* features from the current frame using the Deformable Transformer Encoder and applying a dedicated Deformable Transformer Decoder to process track queries. Specifically, the track queries from the previous frame, $Q_{\text{trk}}^{t-1}$, are updated by the Decoder through interaction with the shared encoder memory $M^t$ of the current frame. Formally, MAT updates the queries as:

$$U_{\text{trk}}^{t-1} = Q_{\text{trk}}^{t-1} + \text{CrossAtt}(\text{SelfAtt}(Q_{\text{trk}}^{t-1}), M^t), \qquad (1)$$

where $\text{SelfAtt}(\cdot)$ refines the query features internally and $\text{CrossAtt}(\cdot, M^t)$ aligns them with the current frame's encoder memory. The MAT module is supervised with a trajectory loss and trained end-to-end alongside the detection component. As illustrated on the left side of Figure 3, this trajectory loss is computed across the entire sequence of length $S$, providing dense supervision for track queries. Such sequence-level guidance allows MAT to effectively distinguish trajectories corresponding to different objects. Benefiting from the attention mechanism, each track query acquires more discriminative features, which enhances the performance of subsequent Transformer Decoder layers. To provide a complete representation, MAT predicts not only the bounding box center coordinates but also its width and height, denoted as $[x, y, w, h]$.

$$\mathcal{L}_{\text{traj}} = \frac{1}{N} \sum^B \sum^S \sum^{N_{\text{trk}}} \text{L1}(\tilde{Y}_{\text{bbox}}, Y_{\text{bbox}}), \qquad (2)$$

where $B$ is the batch size, $N_{\text{trk}}$ is the number of trackers per batch, and $N$ denotes the total number of trackers. $\tilde{Y}_{\text{bbox}}$ and $Y_{\text{bbox}}$ represent the predicted and ground-truth bounding boxes, respectively.

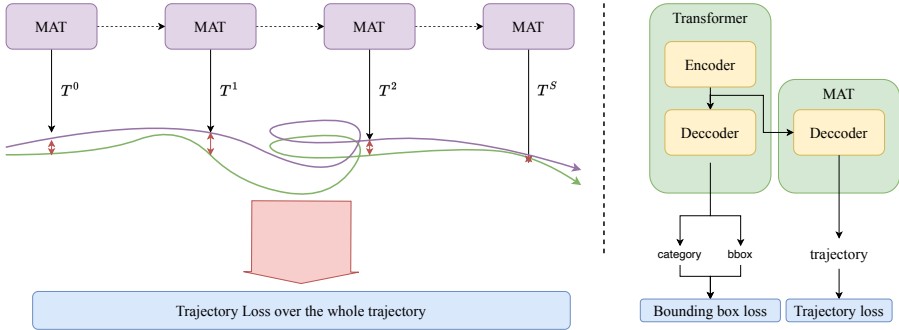

Figure 3: Illustration of the MATR loss design. Left: MAT loss, where dashed lines denote intermediate modules. Trajectory loss is computed across an entire sequence of length $S$, using L1 distance between predicted and ground-truth trajectories. Right: overview of MATR outputs and losses during training.

We deliberately adopt L1 loss instead of IoU-based alternatives since L1 loss directly penalizes deviations in both position and scale, which is crucial for synchronizing feature and positional embeddings. Instead, IoU-based losses focus more on bounding box alone not trajectory.

### 3.4 INFERENCE DETAILS

The inference procedure largely follows MOTR Zeng et al. (2022), with adaptations to accommodate MAT. Since ground truth is unavailable during inference, detection queries are not filtered before being processed by MAT. To handle occlusions, we filter the outputs of the Transformer Decoder. If the confidence score of a tracked object falls below the threshold $\tau_{\text{trk}}$ in the current frame, we retain its embedding temporarily as an inactive trajectory. If its confidence remains below the threshold for more than $\mathcal{T}_{\text{miss}}$ consecutive frames, the inactive trajectory is permanently removed. Similarly, detection results are discarded if their confidence falls below $\tau_{\text{det}}$. In all experiments, we set the thresholds to $\tau_{\text{det}} = 0.7$, $\tau_{\text{trk}} = 0.5$, and $\mathcal{T}_{\text{miss}} = 25$, unless otherwise specified. All other aspects of inference remain consistent with the training procedure, ensuring that MAT can be deployed seamlessly without additional heuristics.

## 4 EXPERIMENTS

### 4.1 DATASET AND METRICS

We evaluate our approach on three challenging benchmarks: DanceTrack Sun et al. (2022), SportsMOT Cui et al. (2023), and BDD100k Yu et al. (2020). **DanceTrack** is a large-scale dataset specifically designed for multi-human tracking in dance scenarios. It is characterized by uniform appearances and diverse, complex motions, which make object association across frames highly challenging. The dataset contains 100 videos, divided into 40 training, 25 validation, and 35 test sequences. Due to its challenging poses and association difficulty, DanceTrack has become one of the key benchmarks for evaluating end-to-end MOT methods. **SportsMOT** is a large-scale dataset densely annotated with every player on the court. It contains 240 videos with more than 150K frames, split into 45 training, 45 validation, and 150 test sequences. The dataset includes a wide variety of dynamic scenes with both moving players and camera motion. A key characteristic is that the test set is substantially larger than the training set, which we leverage to complement the limitations of DanceTrack in terms of scale and diversity. **BDD100k** is a large-scale driving video dataset consisting of 100K videos. Its tracking subset includes 1,400 training, 200 validation, and 400 test videos covering 8 valid categories. BDD100k is the largest and most challenging MOT dataset to date, with highly diverse scenarios in terms of object scales, weather conditions, and times of day. We employ this dataset to comprehensively evaluate the generalization ability of our method, particularly in multi-category settings.

For evaluation, we primarily use the Higher Order Tracking Accuracy (HOTA) metric Luiten et al. (2020), which provides a balanced assessment of tracking by decomposing results into detection accuracy (DetA) and association accuracy (AssA). We additionally report the widely used MOTA and IDF1 metrics to give a broader view of detection and identification performance. For multi-category scenarios, we adopt Track Every Thing Accuracy (TETA) Li et al. (2022), which includes sub-metrics for localization accuracy (LocA) and association accuracy (AssocA). TETA has been

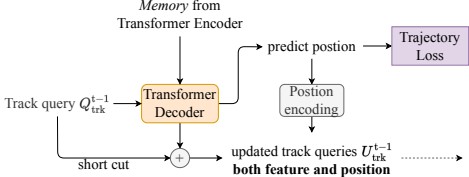

Figure 4: Design of the proposed Motion-Aware Transformer (MAT) module. An additional Transformer Decoder predicts and updates the track queries from the previous frame $Q_{\mathrm{trk}}^{t-1}$, generating refined queries $U_{\mathrm{trk}}^{t-1}$. The MAT module is supervised with a trajectory loss and trained jointly with the main Decoder outputs.

shown to provide a more reliable and comprehensive evaluation than traditional metrics, especially on large-scale multi-class datasets such as BDD100k.

## 4.2 IMPLEMENTATION DETAILS

We build MATR upon MOTR Zeng et al. (2022) and employ several data augmentation strategies, including random resizing and random cropping. The shorter side of each input image is resized to 800 pixels, with the longer side restricted to a maximum of 1536 pixels. Unless otherwise specified, we use ResNet50 as the backbone network. Additionally, we evaluate a larger backbone, Swin-Tiny Liu et al. (2021), to provide a more comprehensive comparison. For hyperparameters, the number of detection queries $N_{\mathrm{det}}$ is set to 300, the sequence length $S$ is fixed at 5, and the dropout probability for track queries is 0.1. The batch size is 1 per GPU, with each batch containing a video clip of $S$ frames sampled at random intervals from 1 to 10 frames. We adopt the AdamW optimizer, using an initial learning rate of $2 \times 10^{-4}$ for ResNet50 and $1 \times 10^{-4}$ for Swin-Tiny. During training, tracked targets with IoU scores below the threshold $\tau_{\mathrm{IoU}} = 0.5$ are filtered out. These training settings remain consistent across all experiments. For the CrowdHuman dataset, we follow CenterTrack Zhou et al. (2020) and apply random spatial shifts to generate pseudo trajectories when including it in training. The overall MATR loss is defined as:

$$\mathcal{L}_{\mathrm{MATR}} = \tau_{\mathrm{traj}}\mathcal{L}_{\mathrm{traj}} + \mathcal{L}_{\mathrm{MOTR}}, \tag{3}$$

where $\tau_{\mathrm{traj}} = 5$ is a fixed weighting coefficient, and $\mathcal{L}_{\mathrm{MOTR}}$ is retained exactly as in MOTR. Our strengthened baseline results are reported in Table 1a and serve as the foundation for our experiments. All ablation studies are conducted on the DanceTrack dataset. Final results are reported on the test sets of DanceTrack and SportsMOT, and on the validation set of BDD100k. Training is performed on 8 NVIDIA A5000 GPUs (24GB each). The ablation studies span 5 epochs, requiring approximately 1 day of training. Detailed ablation results are discussed in Section 4.3. The final MATR model uses the MAT module with one Decoder layer. MATR introduces negligible computational overhead compared to MOTR: +1M parameters and +5% FLOPs, while improving HOTA by more than 9 points. In contrast, MOTRv2/v3 require ¿2x parameters and higher runtime cost, highlighting the efficiency of our design. Following standard practice in MOTR, CO-MOT, and MeMOTR Zhang et al. (2023); Yu et al. (2023); Yan et al. (2023), we combine DanceTrack training data with CrowdHuman. Training lasts 20 epochs with a learning rate decay of 0.1 applied every 8 epochs, requiring about 2.5 days in total. For SportsMOT, training follows the same setup as Dance-Track but excludes additional datasets. For BDD100k, the reduced input size and classification head modification match the settings used in prior end-to-end baselines, ensuring comparability. We also follow the SportsMOT setup, except training runs for 13 epochs with a learning rate decay at epoch 11. Training on BDD100k takes approximately 7 days.

Table 1: Ablation studies on MATR under different components and settings.

(a) Effect of MATR components. Bl-IMP denotes baseline improvement; MAT denotes inclusion of the MAT module.

| Bl-IMP | MAT | HOTA | DetA | AssA | MOTA | IDF1 |
|--------|-----|------|------|------|------|------|
| MOTR | | 54.2 | 73.5 | 40.2 | 79.7 | 51.5 |
| ✓ | | 58.8 | 75.9 | 45.2 | 84.9 | 59.2 |
| ✓ | ✓ | **63.6** | **77.1** | **52.7** | **86.4** | **66.4** |

(b) Effect of the number of Decoder layers in MAT. "Dec" indicates the number of Decoder layers. "KLF" refers to Kalman Filter.

| Dec | HOTA | DetA | AssA | MOTA | IDF1 |
|-----|------|------|------|------|------|
| KLF | 59.6 | 72.9 | 49.5 | 82.2 | 59.0 |
| 1 | **63.6** | 77.1 | **52.7** | **86.4** | **66.4** |
| 3 | 63.5 | **77.9** | 51.8 | 85.1 | 65.7 |

Table 2: Comparison with state-of-the-art methods on the DanceTrack test set. "E2E" indicates whether the method is fully end-to-end. "Params" denotes the total number of parameters in millions (M). Higher values are better for all metrics. ‡ indicates results obtained by including DanceTrack validation set. The MATR configuration corresponds to the final model described in Section 4.2.

| Methods | Attributes | | Metrics | | | | |
|---|---|---|---|---|---|---|---|
| | E2E | Params(M) | HOTA | DetA | AssA | MOTA | IDF1 |
| FairMOT Zhang et al. (2021) | ✗ | / | 39.7 | 66.7 | 23.8 | 82.2 | 40.8 |
| CenterTrack Zhou et al. (2020) | ✗ | / | 41.8 | 78.1 | 22.6 | 86.8 | 35.7 |
| TraDeS Wu et al. (2021) | ✗ | / | 43.3 | 74.5 | 25.4 | 86.2 | 41.2 |
| ByteTrack Zhang et al. (2022) | ✗ | / | 47.7 | 71.0 | 32.1 | 89.6 | 53.9 |
| GTR Zhou et al. (2022) | ✗ | / | 48.0 | 72.5 | 31.9 | 84.7 | 50.3 |
| QDTrack Pang et al. (2021) | ✗ | / | 54.2 | 80.1 | 36.8 | 87.7 | 50.4 |
| OC-SORT Cao et al. (2023) | ✗ | / | 55.1 | 80.3 | 38.3 | **92.0** | 54.6 |
| C-BIoU Yang et al. (2023) | ✗ | / | 60.6 | 81.3 | 45.4 | 91.6 | 61.6 |
| TransTrack (Sun et al., 2020) | ✗ | / | 45.5 | 75.9 | 27.5 | 88.4 | 45.2 |
| MOTR Zeng et al. (2022) | ✓ | 40 | 54.2 | 73.5 | 40.2 | 79.7 | 51.5 |
| MOTRv3-R50 (Yu et al., 2023) | ✓ | 40 | 63.9 | 76.7 | 53.5 | 86.8 | 67.2 |
| MeMOTR Gao & Wang (2023) | ✓ | 48 | 68.5 | 80.5 | 58.4 | 89.9 | 71.2 |
| CO-MOT (Yan et al., 2023) | ✓ | 40 | **69.4** | **82.1** | 58.9 | 91.2 | 71.9 |
| MATR-R50(**ours**) | ✓ | 42 | **69.4** | 81.5 | **59.1** | 91.0 | **72.6** |
| MOTRv2 Zhang et al. (2023) | ✗ | 94 | 69.9 | 83.0 | 59.0 | 91.9 | 71.7 |
| MOTRv3 (Yu et al., 2023) | ✓ | 103 | 70.4 | **83.8** | 59.3 | **92.9** | 72.3 |
| MATR(**ours**) | ✓ | 43 | **71.3** | 82.6 | **61.6** | 91.9 | **75.3** |
| MATR‡(**ours**) | ✓ | 43 | 73.9 | 84.5 | 64.8 | 92.9 | 76.5 |

## 4.3 ABLATION STUDY

**MAT** is a novel module designed to replace the QIM Zeng et al. (2022) module used in MOTR and MOTRv2. Unlike QIM, MAT leverages additional input from the Transformer Encoder and applies direct supervision to the predicted trajectories. To minimize the risk of overfitting, all ablation studies are conducted on the DanceTrack dataset, as it provides extensive training data and highly challenging association scenarios that clearly demonstrate the effectiveness of our method. The ablation results are summarized in Table 1a. Baseline improvements alone already increase HOTA by 4.6 points compared to MOTR. When replacing QIM with the proposed MAT module (with one Decoder layer), we achieve consistent and substantial improvements across all evaluation metrics. Specifically, compared to MOTR, HOTA improves by 9.4 points and MOTA by 6.7 points. Detection accuracy (DetA) increases modestly by 3.6 points, while the association metrics (AssA and IDF1) improve significantly, by 12.5 and 14.9 points, respectively. Because MOT performance is particularly sensitive to association quality, these results underscore the importance of MAT in enhancing tracking stability. Since MAT predicts future positions in advance, it shares some conceptual similarity with the Kalman Filter (Welch & Bishop, 1995) (KLF), a classic prediction mechanism used in traditional tracking-by-detection (TyD) methods. We therefore include KLF as a baseline in Table 1b. The results indicate that while association accuracy (AssA) increases whenever a prediction mechanism is applied, detection accuracy (DetA) drops sharply when using KLF, indicating that its poor prediction quality prevents high-quality detections from being matched. This shows that KLF's prediction accuracy is insufficient for end-to-end MOT. The same limitation has long been observed in tracking-by-detection pipelines: their methods typically achieve higher detection accuracy, but the low-quality predictions from KLF often prevent high-quality detections from being matched during association, ultimately degrading performance. In end-to-end methods, this issue is further magnified - poor predictions from KLF directly propagate into the Transformer Decoder, causing a substantial decline in detection accuracy. We conclude that a linear predictor such as KLF is not suitable for end-to-end architectures, where a learnable predictor capable of being optimized over full trajectories is far more effective. Unlike the Kalman Filter, which is a fixed linear predictor external to the model, MAT learns motion representations jointly with detection in an end-to-end manner. Crucially, it operates directly on Transformer queries, enabling simultaneous optimization of both features and positional embeddings, which classical filters cannot provide. Finally, we investigate the optimal number of Decoder layers within the MAT module. Our initial implementation employs a single Decoder layer, but we also experimented with increasing this to three layers. The results show that although DetA improves slightly by 0.8 points with three layers, all other metrics decrease. Therefore, we adopt a single Decoder layer in MAT for all final experiments, as this design achieves the best balance between accuracy and efficiency.

Table 3: Comparison with state-of-the-art methods.

(a) Results on the SportsMOT test set.

| Methods | HOTA | DetA | AssA | MOTA | IDF1 |
|---|---|---|---|---|---|
| *w/o extra data:* | | | | | |
| FairMOT | 49.3 | 70.2 | 34.7 | 86.4 | 53.5 |
| QDTrack | 60.4 | 77.5 | 47.2 | 90.1 | 62.3 |
| ByteTrack | 62.1 | 76.5 | 50.5 | 93.4 | 69.1 |
| OC-SORT | 68.1 | 84.8 | 54.8 | 93.4 | 68.0 |
| MeMOTR | 70.0 | 83.1 | 59.1 | 91.5 | 71.4 |
| MATR (ours) | **72.7** | **85.3** | **62.0** | **95.1** | **74.6** |
| *with extra data:* | | | | | |
| GTR | 54.5 | 64.8 | 45.9 | 67.9 | 55.8 |
| CenterTrack | 62.7 | 82.1 | 48.0 | 90.8 | 60.0 |
| ByteTrack | 68.9 | 72.1 | 51.2 | 94.1 | 69.4 |
| TransTrack | 68.8 | 82.7 | 57.5 | 92.6 | 71.5 |
| OC-SORT | 71.9 | 86.4 | 59.8 | 94.5 | 72.2 |

(b) Results on the BDD100k validation set.

| Methods | mTETA | mHOTA | mLocA | mAssocA | mAssA |
|---|---|---|---|---|---|
| QDTrack | 47.8 | / | 45.9 | 48.5 | / |
| DeepSORT | 48.0 | / | 46.4 | 46.7 | / |
| MOTR | 50.7 | 37.0 | 35.8 | 51.0 | 47.3 |
| TETer | 50.8 | / | **47.2** | 52.9 | / |
| CO-MOT | 52.8 | / | 38.7 | 56.2 | / |
| MeMOTR | 53.6 | 40.4 | 38.1 | 56.7 | 52.0 |
| MATR (ours) | **54.7** | **41.6** | 41.8 | **59.0** | **53.0** |

## 4.4 COMPARISON ON DANCETRACK

MATR delivers significant improvements in both association accuracy and overall tracking performance. Table 2 reports a comparison between MATR and other state-of-the-art (SOTA) methods on the DanceTrack test set. Our method achieves a new SOTA HOTA score of 71.3, with particularly strong association metrics, including 61.6 AssA and 75.3 IDF1. Due to the inherent limitations of linear motion prediction, tracking-by-detection methods often excel at detection accuracy but fall short in association. For example, OC-SORT Cao et al. (2023) achieves excellent detection accuracy (DetA = 80.3) but struggles with association in complex motion scenarios (AssA = 38.3). In contrast, MATR improves both detection and association metrics simultaneously, highlighting the benefit of explicitly modeling motion to reduce query collisions. For fairness, we also report MATR results with a ResNet50 backbone, denoted as MATR-R50, aligned with MOTR, MeMOTR Gao & Wang (2023), and CO-MOT Yan et al. (2023). We additionally include MOTRv3-R50 (Yu et al., 2023) for reference. Compared with other ResNet50-based methods, MATR achieves a HOTA score of 69.4, matching CO-MOT. Notably: 1) MATR improves HOTA by 15.2 points over MOTR, together with substantial gains across all other metrics. 2) MATR surpasses MeMOTR by 0.9 HOTA and consistently achieves stronger association performance, without relying on merging strategies across historical frames. 3) MATR matches CO-MOT's HOTA performance despite not employing explicit detection enhancements, trailing slightly in DetA (-0.6) but outperforming in AssA (+0.2) and IDF1 (+0.7). These findings demonstrate that the consistent training strategies introduced by MAT not only boost detection accuracy but also yield even greater improvements in association quality. We further compare MATR with larger and more complex models such as MOTRv2 Zhang et al. (2023) and MOTRv3 Yu et al. (2023). MATR achieves a HOTA of 71.3, surpassing MOTRv3 by 0.9 points, while maintaining an elegant and concise end-to-end structure. Despite their complexity, these methods rely on larger backbones (94M and 103M parameters, respectively), whereas MATR achieves superior performance with a smaller Swin-Tiny backbone (43M parameters). Although MATR trails MOTRv3 slightly in DetA (-1.2) and MOTA (-1.0) due to the absence of specialized detection enhancement techniques, it achieves SOTA association results (+2.3 AssA and +3.0 IDF1). This confirms the robustness and effectiveness of explicitly addressing query collisions. In summary, MATR establishes that association accuracy and HOTA remain the central challenges in MOT.

## 4.5 COMPARISON ON SPORTSMOT

Table 3a presents comparisons between MATR and SOTA methods on the SportsMOT test set, all under settings without additional datasets for pretraining or fine-tuning. Detection performance on SportsMOT is generally strong across methods, as indicated by MOTA scores consistently above 90. However, association accuracy (AssA) remains comparatively limited, typically around or below 60. Since SportsMOT already provides a sufficient amount of training data, we emphasize improving association accuracy rather than relying on external datasets. MATR significantly improves upon previous methods across all evaluation metrics. In particular, MATR achieves a new SOTA HOTA score of 72.7, representing gains of 2.7 points over MeMOTR Gao & Wang (2023) and 4.6 points over OC-SORT Cao et al. (2023). While our detection accuracy (DetA) improves only slightly compared to OC-SORT (+0.5 points), the most notable gains are observed in association metrics: +2.9 AssA and +3.2 IDF1 compared to MeMOTR. These results highlight the effectiveness of MATR's

Figure 5: Qualitative comparison between MOTR and MATR. Case A shows that MATR successfully tracks a background object despite occlusion, while MOTR fails due to confusion from a foreground object. Cases B and C demonstrate that MATR maintains accurate tracking despite identity switches caused by crossing trajectories, which MOTR cannot resolve. Case D illustrates that MATR robustly tracks multiple objects in a complex scenario, while MOTR suffers from identity switches and lost trajectories after 30 frames. Note that while MATR reduces most query collisions, it can still fail under extreme occlusions or abrupt camera motion.

motion-aware strategy, which consistently optimizes both tracking and detection queries, leading to substantial improvements in association performance while preserving strong detection accuracy.

### 4.6 COMPARISON ON BDD100K

As the largest and most diverse MOT dataset, BDD100k Yu et al. (2020) poses substantial challenges for tracking methods. Recently, TETer Li et al. (2022) introduced the TETA metric, a comprehensive evaluation framework designed specifically for BDD100k, along with their method. Although TETer achieves the highest localization accuracy (mLocA = 47.2), like other tracking-by-detection methods, it struggles with association metrics, which reduces its overall performance on mTETA and mHOTA. For fair comparison with other end-to-end methods, we adopt a smaller input resolution: the shorter side of the image is resized to 800 pixels, with the longer side limited to 1333 pixels. We also modify the classification head to output 8 categories, corresponding to the valid classes in BDD100k. All other components of our model remain unchanged, and we only use the original training set. During evaluation, we set thresholds to $\tau_{det} = 0.5$, $\tau_{trk} = 0.5$, and $\mathcal{T}_{miss} = 10$. Due to the high computational cost of training (more than 7 days on 8 GPUs), we do not reproduce other methods, and we exclude those that rely on external data. As shown in Table 3b, MATR achieves SOTA performance on nearly all metrics. Compared to MOTR, MATR provides large improvements across the board, demonstrating the effectiveness of our MAT module over the original QIM design. For prior SOTA methods such as CO-MOT Yan et al. (2023) and MeMOTR Gao & Wang (2023), which rely on detection-specific tricks or historical frame information, MATR also achieves superior results. Although CO-MOT and MeMOTR both reach similar association scores (56.2 and 56.7 mAssocA, respectively), our method improves by about 3 absolute points, establishing a new state of the art at 59.0 mAssocA. While MATR does not achieve the top score in localization accuracy, it remains the strongest end-to-end method on mLocA, improving by 3 points compared to CO-MOT. This provides strong evidence that addressing query-type collisions is more critical for overall performance. Finally, MATR sets new SOTA results on the primary evaluation metrics, achieving 54.7 mTETA (+1.1) and 41.6 mHOTA (+1.2). These results further validate the design of the MAT module and its ability to handle query collisions effectively, demonstrating strong generalization from single-class human tracking tasks to large-scale, multi-category tracking scenarios.

## 5 CONCLUSION

We presented the Motion-aware Transformer (MATR), a new approach designed to address key challenges in multi-object tracking by explicitly mitigating query collisions. While prior work has primarily emphasized improving detection accuracy, we argue that within an end-to-end joint optimization paradigm, the optimization of tracking is equally essential and cannot be overlooked. By combining baseline improvements with motion prediction, MATR achieves state-of-the-art performance on the DanceTrack, SportsMOT, and BDD100k datasets. These results demonstrate that end-to-end frameworks can be effectively optimized for complex multi-object tracking tasks, providing both efficiency and scalability. It is important to note, however, that MATR mitigates query collisions through motion prediction but does not completely eliminate them. A fundamental challenge remains: how to decouple tracking and detection components in a way that preserves the elegance of an end-to-end framework. Such a decomposition could make it possible to fully remove query collisions while optimizing the two components separately, potentially leading to further performance improvements. We leave this as an exciting direction for future work. In summary, MATR shows that explicitly modeling motion within transformers is a simple yet powerful principle for advancing end-to-end multi-object tracking.

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
