# OpenReview forum: "Motion-Aware Transformer for Multi-Object Tracking"
_ICLR.cc/2026/Conference — ICLR 2026 Conference Withdrawn Submission_

### Official Review · Reviewer_ABx3 · 2025-10-30

**Soundness:** 2
**Presentation:** 3
**Contribution:** 2
**Rating:** 6
**Confidence:** 3

**Summary:**

This paper introduces the Motion-Aware Transformer (MATR), a new approach for multi-object tracking (MOT) in videos. The method aims to solve the query collision problem that occurs in current DETR models during the detection and tracking processes. MATR explicitly predicts the motion of objects across frames and updates the track queries in advance, reducing query drift and improving the consistency of tracking and detection. Extensive evaluations on multiple benchmark datasets show that the method achieves state-of-the-art performance in association accuracy and overall tracking results.

**Strengths:**

1. MATR introduces the innovative concept of motion-aware query updates within the Transformer framework, alleviating query collision issues common in traditional MOT models.

2. The method demonstrates significant improvements in tracking and detection accuracy, particularly in challenging crowded or fast-moving scenarios. The ablation studies clearly showcase the contributions of each component, validating the effectiveness of motion prediction.

3. The paper is well-written, effectively explaining the motivation behind the proposed method, its design, and the experimental setup. Figures and diagrams help in visualizing the concepts.

4. MATR achieves state-of-the-art results on several challenging datasets (DanceTrack, SportsMOT, and BDD100k), proving its potential for real-world applications in multi-object tracking.

**Weaknesses:**

1. Although MATR significantly reduces query collisions, it does not fully eliminate them. Future work should explore how to decouple the tracking and detection components for even greater performance improvements.

2. While the method claims to predict motion, there is no direct proof presented, such as visualizations or attention analysis, to demonstrate this ability.

3. The paper could have benefited from considering more diverse aspects of the problem or presenting stronger, more solid contributions in terms of novelty.

**Questions:**

1. How does the model perform when tracking objects with highly irregular motion or extreme occlusions? Could you provide visual results to support this?

2. Would the authors consider open-sourcing the method? Also, as the paper mentions that query collisions are reduced but not fully eliminated, what are the next steps to address this issue and how would it impact the overall model design?

3. The paper does not provide direct evidence for motion prediction, such as visualizations or attention analysis. Can these be provided to further support the claim?

---

### Official Review · Reviewer_dYk4 · 2025-11-01

**Soundness:** 3
**Presentation:** 3
**Contribution:** 2
**Rating:** 4
**Confidence:** 4

**Summary:**

This paper proposes a new method for multi-object tracking. It improves the transformer-based framework using a separate decoder for trajectory prediction. It proposes a new module to explicitly predict object movement across frames and update track queries. Experiments on public MOT benchmarks demonstrate the effectiveness of the proposed method.

**Strengths:**

1. The idea of predicting trajectory using another decoder makes sense.
2. The implementation of the MAT module is reasonable.
3. The proposed method achieves leading performance in multiple public benchmarks.

**Weaknesses:**

The main concern is the limited methodology contribution. Although using another decoder to predict trajectory and update queries is effective, the technical part is straightforward and analysis is not comprehensive enough. Overall, it has practical value, but is more application-oriented. Hence I tend not to accept this version.

**Questions:**

Will the MAT module also work for other transformer-based models, besides MOTR? Experiments are other transformer-based models are encouragred, which can demonstrate the universality of the proposed method.

---

### Official Review · Reviewer_GqHZ · 2025-11-08

**Soundness:** 2
**Presentation:** 1
**Contribution:** 2
**Rating:** 2
**Confidence:** 4

**Summary:**

This paper introduces a novel approach for multi-object tracking (MOT) by integrating motion-aware mechanisms into a DETR-based framework. The authors raise the issue of query collision and propose a core contribution that explicitly incorporates motion learning into the model, which aims to improve the accuracy and robustness of tracking multiple objects. Experimental results demonstrate the effectiveness of the proposed method. However, there remains a significant gap between the current quality of this paper and the standards required for acceptance.

**Strengths:**

The paper outlines the problem of query collision, which is an interesting issue existing in DETR-family frameworks for multi-object tracking problem, and introduces a novel motion-aware mechanism integrated into a DETR-based framework, alleviating the afore-mentioned issue.

**Weaknesses:**

1. Clarity and presentation issues (Major revision is required). The manuscript requires substantial editing for clarity and professional presentation. These issues suggest the work was not thoroughly proofread or revised before submission. The overall writing quality is poor, which significantly hinders readability. There are numerous basic formatting errors throughout the text. Examples include incorrect citation formats throughout, visible artifacts (e.g., the redundant "IOU(A)" label in Figure 1b, the stray question mark in line 308), and suboptimal presentation of data (e.g., poor formatting in Table 3 and Figure 5).

2. Ambiguity in core motivation and problem definition The paper's central argument regarding the "query collision issue" is conceptually weak and lacks precise clarity, diminishing the motivation for the proposed solution. The introduction intends to establish a unique issue—the disharmony between track queries and detection queries in the unified decoder. However, the subsequent analysis and discussion often blur this distinction, making the phenomenon appear to be merely a consequence of target occlusion and crowded scenes (i.e., standard MOT challenges) rather than a novel architectural flaw specific to the DETR-MOT design.

3. Technical and robustness limitations of the proposed method. The proposed method exhibits several technical limitations and robustness concerns. First, the query updating mechanism appears insufficient for handling prolonged target occlusion, and the proximity of similar objects introduces a high risk of track query contamination. Second, the system's reliance on explicit motion prediction makes it vulnerable to non-linear or irregular motion trajectories, where the module may introduce more interference than benefit. Finally, the authors' heavy dependence on artificially introduced thresholds during inference significantly compromises the model's generalizability and robustness against diverse, real-world scenarios.

**Questions:**

Please see the questions listed in Weaknesses part.

---

### Official Review · Reviewer_Z2Ja · 2025-11-08

**Soundness:** 3
**Presentation:** 3
**Contribution:** 3
**Rating:** 6
**Confidence:** 3

**Summary:**

This paper addresses a challenge of multi-object tracking (MOT) in video sequences, particularly concerning query conflicts in existing end-to-end Transformer-based models. The core innovation is the proposed Motion-Aware Transformer (MATR), which explicitly predicts the object's movement between frames to dynamically update and refine the track queries before the main cross-attention steps. This prospective query refinement significantly mitigates the interference between simultaneous detection and tracking tasks, leading to more stable and accurate object association. The authors demonstrate strong performance across several public MOT benchmarks, showing that MATR achieves state-of-the-art or highly competitive results, effectively validating the framework's effectiveness.

**Strengths:**

1.	The strategy of leveraging predicted target motion to proactively update the track queries is novel in the context of end-to-end MOT, offering a simple yet highly effective mechanism for improved temporal consistency and collision avoidance among queries.

2.	The paper provides a comprehensive and systematic survey and discussion of current end-to-end Multi-Object Tracking frameworks.

3.	The experimental section demonstrates notable performance gains over strong baseline methods and achieving state-of-the-art or highly competitive results on several challenging datasets like DanceTrack and BDD100k.

4.	The ablation study, like Table 1, is clearly presented, offering quantitative insight into the individual contribution.

**Weaknesses:**

1.	The current motion prediction relies on simple bounding box regression. The authors should discuss or explore the potential benefits and tradeoffs of integrating other external motion models, such as a Kalman Filter or deep motion models. In addition, the Related Work section should include a more comprehensive and dedicated discussion of prior work (e.g. [*]) utilizing explicit motion prediction techniques.

2.	The qualitative results could be significantly enhanced. Currently, the visualizations (e.g., Figure 5) are too small, making it difficult to discern subtle details of the tracking results.


[*] Standing Between Past and Future: Spatio-Temporal Modeling for Multi-Camera 3D Multi-Object Tracking, CVPR2023.

**Questions:**

N/A

---

### Note · Authors · 2025-11-13

I have read and agree with the venue's withdrawal policy on behalf of myself and my co-authors.